# Low-Cost Sensor Deployment on a Public Minibus in Fukushima Prefecture

**DOI:** 10.3390/s24051375

**Published:** 2024-02-21

**Authors:** Rakotovao Lovanantenaina Omega, Yo Ishigaki, Sidik Permana, Yoshinori Matsumoto, Kayoko Yamamoto, Katsumi Shozugawa, Mayumi Hori

**Affiliations:** 1Graduate Program in Nuclear Science and Engineering, Faculty of Mathematics and Natural Sciences, Bandung Institute of Technology, Bandung City 40132, Indonesia; 2Research Center for Realizing Sustainable Societies, The University of Electro-Communications, Tokyo 182-8585, Japan; 3Department of Applied Physics and Physico-Informatics, Faculty of Science and Technology, Keio University, Yokohama 223-8522, Japan; 4Graduate School of Information Science and Technology Department of Informatics, The University of Electro-Communications, Tokyo 182-8585, Japan; 5Graduate School of Arts and Sciences, The University of Tokyo, Tokyo 153-8902, Japan; 6College of Arts and Sciences, The University of Tokyo, Tokyo 153-8902, Japan

**Keywords:** public bus, POKEGA, IoT, ambient dose, half-life

## Abstract

This study analyzed radiation dose data to observe the annual decline in ambient radiation doses and assess the factors contributing to fluctuations in reconstructed areas of the Fukushima prefecture. Utilizing a novel mobile monitoring system installed on a community minibus, the study employed a cost-effective sensor, namely, Pocket Geiger which was integrated with a microcontroller and telecommunication system for data transfer, access, visualization, and accumulation. The study area included the region between Okuma and Tomioka towns. The ambient dose rate recorded along the minibus route was depicted on a map, averaged within a 1 × 1 km mesh created with the Quantum Geographic Information System. To ensure accuracy, the shielding factor of the minibus material is determined to adjust the dose readings. A significant decrease (*p* < 0.001) in the radiation dose ranges from 2022 to 2023 was observed. The land use classification by the Advanced Land Observation Satellite revealed an ecological half-life ranging from 2.41 years to 1 year, suggesting a rapid radiation decay across all land types. This underscores the close connection between radiation attenuation and environmental factors, as well as decontamination efforts across diverse land categories.

## 1. Introduction

The Fukushima Daiichi nuclear accident occurred on March 11, 2011, sustaining damage from a powerful earthquake and tsunami with a magnitude of 9.024, the highest ever recorded in Japan [1]. This catastrophic event led to the release of various radioactive substances, including iodine-131 (^131^I), cesium-137 (^137^Cs), and cesium-134 (^134^Cs), into the environment, notably contaminating the surrounding areas [2]. Consequently, monitoring for radioactive cesium became imperative [3].

Over a decade since the incident, radiocesium, particularly the longer-lived isotope ^137^Cs with a half-life of 30 years, remains the primary concern. Notably, despite being released in comparable quantities, less than 10% of the initially emitted shorter-lived isotope, ^134^Cs (with a half-life of 2 years), was still detectable in the environment in 2022 [4]. Understanding the prolonged presence of radiocesium in the environment is crucial for ongoing monitoring and remediation efforts.

In the aftermath of this knowing accident, residents in the Fukushima prefecture expressed concerns regarding their health. They found reassurance lacking in official Japanese communications and felt that they did not receive adequate information, even though such information should have been shared and accessible to assist them in making informed decisions regarding their situation [5]. The Japan Atomic Energy Agency collected environmental monitoring data on air dose rates, originating from both airborne and deposited radionuclides. Furthermore, the agency gathered information on radionuclide concentrations in various environmental compartments, including air, ground surface, soil, saltwater, marine soil, river water, river sediment, groundwater, drinking water, and food [6]. The Environmental Monitoring Database, publicly accessible through an Internet website, was established utilizing data collected by the Japan Atomic Energy Agency [5]. Various methods, including the Kyoto University Radiation Mapping System, a global positioning system (GPS)-linked automatic radiation measurement system developed by the Kyoto University Research Reactor Institute in response to the Fukushima Daiichi Nuclear Power Station accident, were employed [7]. Other approaches involved strategically positioned sensors, data visualization tools, and data aggregation techniques, and transforming abstract data into actionable insights [8]. These insights were made available to the community, local authorities, and the broader scientific community, enabling a more informed and proactive approach to radiation safety [9]. Although methods such as scintillation detectors are resource-intensive and limited in comprehensive area monitoring, the implementation of an inexpensive system in the Fukushima prefecture addresses these challenges, providing comprehensive coverage, efficient resource usage, and real-time data accessibility.

To achieve cost-effectiveness, the low-cost Pocket Geiger (POKEGA) was developed in August 2011 and field-tested using mobile monitoring [10]. This series of inexpensive mobile radiation detectors was created to satisfy the demand for an affordable radiation monitoring system accessible to the public. The POKEGA sensor was integrated into fixed systems in selected Fukushima prefecture areas [11], utilizing the GPS and networking capabilities of smartphones for logging and data sharing. This approach not only transforms radiation monitoring into a communal effort marked by transparency and collective awareness but also empowers communities and local authorities with tools and insights to ensure the safety of residents in radiation-affected regions.

The innovative use of public transportation as a mobile radiation monitoring system enhances data collection across different areas in the Fukushima prefecture, offering a more comprehensive picture of radiation levels. Unlike using smartphones for sensor utilization, this system employs a microcontroller for increased energy efficiency, drawing power from the minibus and lasting approximately 20 h on battery. This Internet of Things (IoT)-driven mobile monitoring system, that is executed with precision, provides an unprecedented vantage point into the intricate dynamics of ambient dose rates in the region.

This study introduces a novel mobile monitoring system utilizing the POKEGA sensor inside a community minibus connecting the residential area in the Fukushima prefecture towns. This minibus is a BYD J6 2.0 compact electric bus created especially for Japan’s city transportation requirements. It has dimensions of 6.99 m by 2.08 m by 3.06 m and can hold up to 24 passengers. Due to its ability to cover roughly 210 km on a single charge, the bus is a dependable option for community bus services. 

In the context of SAFECAST, citizens perform ambient dose measurements, which could potentially influence the data quality. Moreover, this study requires several thousand measurements to encompass various regions [12]. The advantage of minibus monitoring with the POKEGA system proposed in this study is the use of a single sensor capable of covering the entire route between two towns. 

The public minibus route originates from Okuma, a town completely evacuated after the nuclear disaster, where residents were initially permitted to return only during daylight hours. In April 2019, certain areas of the town were subjected to effective decontamination, leading to the authorization for residents to reoccupy these regions. Subsequently, evacuation orders were lifted in April 2022, and the town was declared free from evacuation in June 2023 [13,14]. This route connects to Tomioka, where the Japanese government lifted the evacuation order in April 2017. However, the reoccupation of residences by previous inhabitants remained restricted and was officially lifted in April 2023. The route encompasses the reconstruction and revitalization area, where evacuation restrictions were eased after decontamination. 

At this stage, continuous monitoring and assessment of these areas are imperative for ensuring the safety and well-being of returning residents and providing insights for ongoing decontamination efforts. This study compared the decline in the ambient dose by examining data from February 2022 to 2023. If a difference was identified, a *t*-test is employed to confirm its significance, and land use is considered to understand the environmental factors affecting the decay of the radioactive component in the minibus route area. The evaluation also includes determining the ecological half-life within the categories of each land use. This comprehensive approach aims to offer a more accurate understanding of changes in ambient radiation levels over time and the influencing factors. Consequently, this contributes to ongoing efforts in radiation protection and environmental monitoring in Okuma and Tomioka Town.

## 2. Material and Methods

### 2.1. Radiation Monitoring System

The primary vision behind creating the POKEGA device was to ensure accessibility for the public, considering both the cost and installation. The version utilized in this study, Type 6, is the latest model specifically designed for remote sensing in open areas. The range of the POKEGA is estimated to be 0.05–10 mSv/h (equivalent to ^137^Cs), covering the spectrum of radiation levels in Japan. Equipped with the GPS, the sensor marks coordinates at each point every second during recording. The sensor was tested at the Fukushima Daiichi nuclear power plants in 2012, demonstrating its ability to read from 2 to 20 µSv/h [10,11].

We developed a POKEGA system in a minibus, at the core of the IoT setup, and pivotal for monitoring radiation levels. A microcontroller interfaces with the sensor to process and collect data, interpreting electrical pulses generated in response to the measurement. 

The POKEGA sensor specifically measures the ambient dose H*(10), as illustrated in Figure 1, which depicts the schematic of the minibus monitoring system. The dose is expressed in microsievert per hour, obtained from a height of 1 m above the ground level. The measurement interval was 30 s, resulting in the recording of two points every minute, and the time constant for the moving average of the POKEGA system was set at 20 min [11]. Consequently, if a minibus travels at 40 km/h, the average air dose over a linear distance of 333 m will be recorded.

At the same time, location tracking utilizes the GPS for added context and precision, enabling the device to provide radiation data with accurate geographic coordinates. Power is supplied steadily via a universal serial minibus (USB) connected to the minibus power for continuous operation, though measurements are stopped when the minibus is not in operation. Radiation monitoring systems leverage long-term evolution (LTE) communication in JavaScript object notation (JSON) protocol for real-time data transfer, facilitating remote monitoring and rapid response to changing radiation levels. 

All collected radiation data, complemented by location information, is transmitted to cloud systems, notably Ambient (Ambient Data Inc., Tokyo, Japan), a Japanese IoT data visualization service employing the JSON protocol. Cloud-based platforms offer real-time monitoring, historical data analysis, and data visualization, accessible through the web. This empowers users to make informed decisions regarding radiation safety and research.

The IoT radiation monitoring system uses the Wio LTE JP Version, a microcontroller module by Seeed Technology Co. Ltd., Shenzhen, China. It is equipped with a Grove connector, an STM32F4 microcontroller, and an LTE module. GPS functionality incorporates the M5StackGPS (M5Stack Technology Co. Ltd., Shenzhen, China) and the GroveGPS from Seeed Technology Co. Ltd. The M5StackGPS supports multiple satellite systems using an AT6558 chip. The GroveGPS, with a SIM28 chip, supports GPS, GLONASS, and Beidou satellites. The Wio Extension-Real-Time Clock (RTC), a supplementary board for the Wio LTE platform, provides RTC functionality through the inter-integrated circuit (I2C) interface, using the NXP PCF8523 chip. It includes a micro-USB port for power, an I2C port for communication, and a USB power output controllable via a switch, enabling it to power the Wio LTE. The data processed by the microcontroller are visualized in the cloud system. The IoT data visualization service, Ambient, efficiently transfers the data generated by the microcontrollers, which are transmitted to the cloud. These readings, accompanied by pertinent details, are acquired, stored, and presented in a user-friendly manner. The graphs and dashboards are designed to be simple, serving as commonly used tools for visualization. The supplied data are accessible in real-time, requiring minimal initial setup for immediate viewing and analysis.

### 2.2. Shielding Factor

Given that the ambient dose measurement occurs inside the bus, the shielding material of the minibus body must be considered. To address this, we conducted a random walk along the minibus route using the same sensor model, connected to a smartphone 1 m above the ground level at the same height as the bus measurement. This dataset allowed us to calculate a ratio, serving as a shielding factor to adjust the radiation dose in this study. Approximately 80 points were collected, with measurements taken every 2 min. Through this experiment, we aimed to determine the outdoor measurement for the year 2023. Another outdoor research initiative was conducted in 2022 and applied to establish the shielding factor within the dose measured in the same year.

The shielding factor is defined as the ratio of the indoor and outdoor ambient dose rate and is calculated as follows [15].
(1)$$S=\frac{D_{in}}{D_{out}}$$
where $D_{in}$ is the ambient dose recorded inside the bus, and $D_{out}$ is the dose recorded during the field survey. The outdoor radiation measurement was conducted in Okuma town, specifically in a delineated area along the minibus route. Simultaneously, indoor radiation levels, reflecting the radiation inside the bus, were recorded within the same geographical scope for more accurate coverage. 

### 2.3. Ambient Dose Mapping Based on Land Use Classification

The implementation of this monitoring system commenced in 2019, with continuous real-time recording and aggregation of data from the sensor network. The minibus route covered Okuma and Tomioka towns, two locations of particular interest owing to their proximity to the Fukushima Daiichi nuclear power plant. The start point of the study area was located at Okuma Town Hall (37.3821551° and 140.9582305°), and the end point was at Tomioka station (37.3352527° and 141.0220414°) within 8.8 km. To assess changes and ensure accurate analysis of ambient radiation levels, we selected two specific time points between February 2022 and 2023. This straightforward approach allows for direct observation of the system’s impact, minimizing potential confounding factors and facilitating a clear comparison. This method also streamlines the data collection and analysis process. This selection enables a comparative study that considers the same seasonal variations and other temporal influences, helping to identify trends and patterns in radiation levels.

The construction of the close map involved the use of geographic information system (GIS) software, specifically quantum (QGIS version 3.34.3). This software overlays the collected radiation data onto the geographical maps of Okuma and Tomioka towns. The data from each point are mean-based and represented within a 1 × 1 km grid as displayed in Figure 2. This choice is relevant because the distance between individual measurement points is approximately 333 m, much less than 1 km.

The bus route in land use classification by Advanced Land Observation Satellite (ALOS) is displayed in Figure 2, utilizing the high-resolution land-use map of Japan. The ALOS High-Resolution Land Use and Land Cover Map was created by the Earth Observation Research Center (EORC) of the Japan Aerospace Exploration Agency (JAXA). This map, derived from satellite Earth observation data, provides details on land use and land cover at regional and national levels. It comes in several versions, including ver. 21.11, ver. 18.03, and ver. 16.09. The map comprises 12 categories, including bamboo forests, bare lands, water bodies, urban regions, paddy fields, croplands, grasslands, deciduous broad-leaf forests, deciduous needle-leaf forests, evergreen broad-leaf forests, and solar panels [16]. For this study, bamboo forests, deciduous broad-leaf forests, deciduous needle-leaf forests, and evergreen broad-leaf forests are merged into one class and categorized as forests.

### 2.4. Ecological Half-Life Decay

The assessment of ecological half-life, representing the combined impact of radioactive decay and environmental processes such as migration, weathering, and organism uptake, signifies the time required for a radioactive substance, such as ^137^Cs, to diminish to half of its original quantity in a specific environment [17]. This assessment carries considerable implications for ambient dose rate monitoring and radiation safety. Understanding the ecological half-life is integral to comprehending the persistence and dynamics of radioactive contaminants in the environment, influencing ambient radiation levels over time. By quantifying the ecological half-life of these contaminants, we gain valuable insights into the duration of their impact on local ecosystems and, consequently, the ambient dose rates in the region [18].

The ecological half-time of the dispersion of radioactive material in the environment is calculated using the following equation [18]:(2)$$T_{eco}=ln2\;\frac{t_{2}-t_{1}}{ln\;(\frac{D_{1}(t_{1})}{D_{2}\;{(t}_{2})})}$$
where $D_{1}$ and $D_{2}$ represent the mean ambient dose rates recorded by the minibus monitoring system at time points $t_{1}$ and $t_{2}$ from February 2022 to 2023. The mean dose rates $D_{1}$ and $D_{2}$ are determined by subtracting the natural background value of 0.04 µSv/h in Fukushima before the nuclear power plant accident. We used Tukey and *t*-test models to identify significant changes with *p*-values less than 0.05 in ambient radiation dose levels across different land uses from 2022 to 2023, providing insights into ambient dose parameter shifts and contributing factors such as ecological half-life. 

## 3. Results and Discussion

### 3.1. Shielding Factor

The calculated shielding factor is summarized in Table 1 resulting from the ratio of indoor and outdoor ambient doses inside the bus. In 2022, the outdoor and indoor average ambient dose levels were 0.30 μSv/h and 0.09 μSv/h, respectively, yielding a shielding factor of 0.30. This indicates that approximately 30% of the outdoor radiation penetrated the minibus in 2022. After one year, in 2023, the average outdoor ambient dose was 0.17 μSv/h, while the indoor ambient level decreased to 0.07 μSv/h. The shielding factor slightly increased to 0.40, indicating that approximately 40% of the outdoor radiation penetrated the bus. The *t*-test revealed that this difference was not statistically significant, prompting consideration of various parameters.

Lauridsen et al. [15] stated that the open area shielding factor for the bus is 0.40 with passengers and 0.35 without passengers. The shielding factors for vehicles passing through areas contaminated by activity released into the atmosphere from a reactor accident and found that modifying factors, such as mutual shielding by nearby buildings and passengers, can affect the shielding factors for ordinary cars and buses in both urban and open areas, as well as areas with single-family houses. This result is consistent with a range of shielding factors for the minibus monitoring system determined in the present study equally 0.30 in 2022 and 0.40 in 2023 as displayed in Table 1. 

Different materials exhibit varying abilities to shield against radiation. The placement of the sensor near passengers could also influence the readings, as human bodies can provide some degree of protection from radiation, particularly for lower-energy gamma rays, and this phenomenon is known as self-shielding [15]. However, the effect would likely be minimal unless the passengers are in proximity to the sensor. We incorporated quantifying the shielding factor through outdoor measurements; however, the temporal constraints imposed on the data collection process were relatively stringent, limiting the comprehensive coverage of the entire minibus route. This limitation might affect the accuracy of the results.

### 3.2. Mapping of Ambient Dose Rate

The dose decreased from February 2022 to 2023, as indicated by the variation in color on the map shown in Figure 3, representing the level of the median dose in each mesh. The fluctuation in values may be attributed to environmental parameters within each land use surface of the recorded coordinates along the minibus route. The color differences from Figure 3a,b are visibly different reflecting the reduction of the dose from the range of 0.15 μSv/h in red to 0.02 μSv/h in dark blue. 

In comparison, the average ambient dose rates in Tomioka town ranged from 0.15 to 0.18 μSv/h indoors between 2018 and 2019 [19]. Following the Fukushima Daiichi nuclear power plant accident, the International Commission on Radiological Protection (ICRP) Publication 146 describes how the long-term phase of a nuclear accident scenario starts when the radiological status is recognized. 

These incidents emphasize the importance of following the Commission’s recommendation and adopting a lower reference standard for long-term phases. For the current exposure scenarios, the reference level should be chosen from the lower half of the suggested range of 1–20 mSv annually, considering the population’s precise dosage distribution and the effects of the exposure scenario on society, the environment, and the economy. This implies that indoor radiation exposure doses have been effectively regulated to align with the public dose limit of 1 mSv per year. Adherence to guidelines set forth by the ICRP ensures that individual residents experiencing continuous or frequent exposure are subject to an annual practical dose limit of 1 mSv [20,21].

To assess the significance of the dose level difference, a *t*-test was conducted as shown in Figure 4. The dose was corrected using the shielding factor to accurately correlate with the ambient dose rate readings obtained in 2022 and 2023.

The ambient radiation dose data of February 2022 and 2023 indicated a noteworthy decline in radiation levels during the latter year. This is substantiated by a significant reduction in the annual mean dose from 0.210 to 0.106 µSv/h. A *t*-test, assuming unequal variances, confirmed this marked difference (*p* < 0.001).

In April 2013, the ambient dose in Tomioka ranged from 0.75 to 4.10 μSv/h; in 2018, the range was approximately 0.08 to 3.00 μSv/h [19]. This historical context is particularly relevant when considering the decrease in ambient dose levels over time. According to the Ministry of the Environment’s designation of priority investigation areas, decontamination implementation plans aimed for a radiation level of 0.23 μSv/h measured between 50 cm and 1 m above the ground [20]. This value was converted and achieved an optimal level of 1 mSv/year [22]. The average dose in this study was below the target of 0.23 μSv/h, suggesting that decontamination efforts have indeed contributed to a decrease in ambient dose rate levels.

The meaningful variance in ambient dose rates between 2022 and 2023 can be attributed to various factors, including the natural decay of radioactive materials, remediation efforts, land use changes, climatic conditions, and statistical variability. These findings provide insights into temporal trends, ecological half-lives, and variations in dose rates across different land use categories. The study suggests the environment is gradually recovering from previous radiation exposure, guiding future environmental monitoring and management strategies. However, the methodology’s dependence on one minibus presents a vulnerability to data collection disruptions, and future studies should consider incorporating measurement apparatus deployment in multiple buses.

The community minibus runs on narrow roads and reflects the amount of radiation from the surrounding land uses. The decontamination process on the Fukushima roads, which spanned a total distance of 1635 km, has been successfully accomplished. This comprehensive process involved the removal of fallen leaves, moss, mud, and other deposits including the edge resulting from the low-level dose in the road. In addition, methods such as brush cleaning and high-pressure water cleaning were implemented [23,24]. For that, these pathways, which are less than 4 m wide, more closely resemble streets than roads and are situated within residential and mountainous areas. Therefore, the recorded values exhibit major variation contingent upon the nature of the surrounding land uses [25]. This variation indicates the natural attenuation occurring in the vicinity of pathways, which are readily accessible to the public.

### 3.3. Ecological Half-Life and Land Use Classification

Table 2 presents the mean ambient dose rates for various land use classifications in February 2022 and 2023, accompanied by corresponding ecological half-life values. The observed decrease in ambient dose rates from February 2022 to 2023 indicated a gradual environmental recovery from prior radiation exposure across diverse land categories, including bare land, crop areas, forests, grasslands, rice paddies, urban areas, and solar panels. Notable variations in radioactive material levels were evident, with bare land exhibiting the longest half-life of 2.4 years and cropland displaying the shortest at 1 year. These findings highlight slight differences in ecological half-lives among land use classifications and are consistent with the study by Kinase et al. [26]. The fractional distributions of the short-term component demonstrate clear dependence on land use.

The statistical analysis reveals changes in ambient dose rates across different land use types in Okuma and Tomioka towns in Fukushima from 2022 to 2023. The details of the difference in 2022 and 2023 are shown in the Appendix A Table A1 and Table A2. Despite decontamination efforts and the natural decay of radioactive materials, the urban areas have significantly different ambient dose rates with *p*-values less than 0.05 from natural such as forest, bare land, and grassland, as well as agricultural lands such as rice paddies and crops. The differences between rice paddy–urban and bare land–urban in 2022 may indicate changes in radiation levels due to remediation efforts, land use changes, and measurement methods. The findings suggest that work and the natural decay of radioactive materials have contributed to reducing ambient dose rates.

The adoption of the comprehensive land use classification system (ALOS), encompassing categories such as bare land, crop, forest, grassland, rice paddy, urban, and solar panels, bolsters the specificity of the results. The temporal decay dynamics emphasize the pronounced reduction in ambient dose rates, especially in urban areas and structures, underscoring the impact of environmental factors. The assumption of a constant ecological half-life and inherent variability in environmental conditions may introduce limitations to the study. Furthermore, the absence of solar panel data for 2022 and reliance on 2023 data may restrict the comprehensive assessment of radiation attenuation trends for this specific land type.

## 4. Conclusions

In this study, we comprehensively analyzed and assessed the factors influencing ambient dose levels in the Fukushima prefecture. The determined shielding factor indicated that approximately 40% of outdoor radiation penetrated the bus. The corrected ambient dose rate between February 2022 and 2023 exhibited a significant difference (*p* < 0.001), measuring less than the decontamination threshold of 0.23 μSv/h [20]. This measurement was securely within the parameters outlined in the ICRP Publication 146, which discussed the long-term phase of a nuclear accident and the present exposure scenarios, suggesting the reference level must be selected from the lower half of the proposed 1–20 mSv range. The remaining radionuclide in the environment displayed an ecological half-life range of 2.41 years to 1 year, affirming the efficacy of decontamination efforts. The slight variation in ambient dose rate attenuation across various land use categories underscored the nuanced impact of environmental factors.

These key findings from ICRP data analysis underscored the valuable contribution of public minibus monitoring to a safe environment and public health. The temporal decay dynamics emphasized a substantial reduction in ambient dose rates, particularly in urban areas and structures. Although the current system provides data accessibility to users, there is room for expansion and enhancement.

The implementation of public minibus monitoring allowed for the comprehensive tracking of radiation level changes over time and under diverse conditions, essential for a thorough understanding of ambient dose levels. In future initiatives, there are plans to integrate a visual interface on public buses providing real-time information on radiation levels. This initiative aims to enhance public safety and notably contribute to increasing awareness of the societal implications of radiation monitoring. Providing easy access to this information for the public fosters a sense of safety and confidence among individuals returning to their residences.

## Figures and Tables

**Figure 1 sensors-24-01375-f001:**
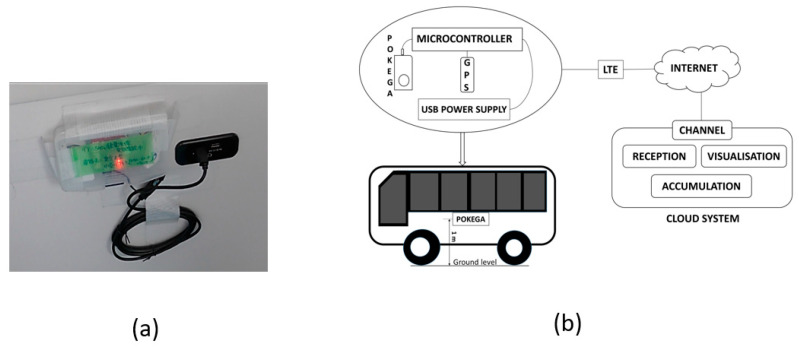
Picture and schematic diagram of the minibus monitoring system. (**a**) Picture of the installation of the sensors inside the minibus and, (**b**) schematic diagram.

**Figure 2 sensors-24-01375-f002:**
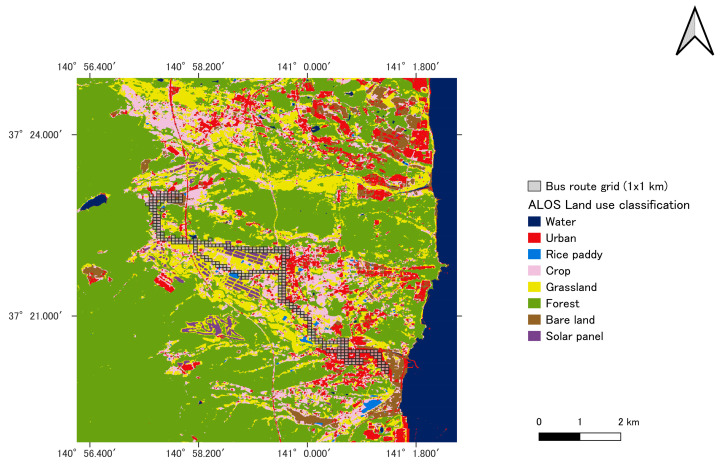
Land use visualization and 1 × 1 km grid of minibus routes. The map is created using QGIS.

**Figure 3 sensors-24-01375-f003:**
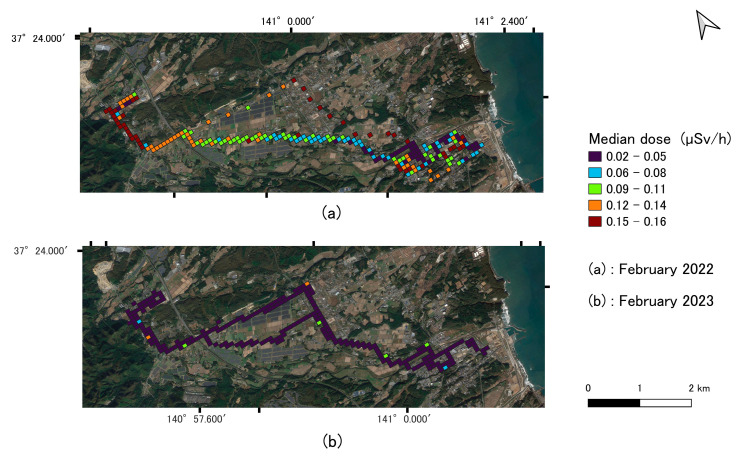
Comparison of indoor ambient dose rate in February 2022 and February 2023. The map is created using QGIS.

**Figure 4 sensors-24-01375-f004:**
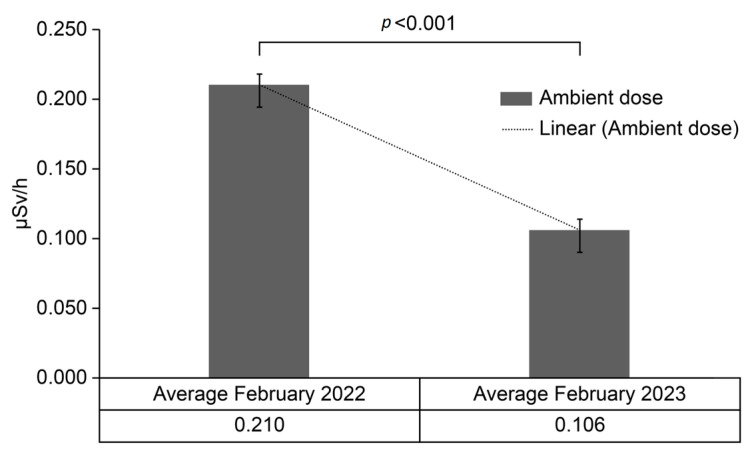
Ambient dose variation from February 2022 to 2023.

**Table 1 sensors-24-01375-t001:** Shielding factor of the minibus body material.

	Mean Outdoor Dose [µSv/h] (Standard Error)	Mean Indoor Dose [µSv/h] (Standard Error)	Shielding Factor (Standard Error)
February 2022	0.30 (0.04)	0.09 (0.02)	0.30 (0.035)
February 2023	0.17 (0.04)	0.07 (0.02)	0.40 (0.035)

**Table 2 sensors-24-01375-t002:** Ambient dose rates are based on the land use classification and ecological half-life.

ALOS Classification	Mean Dose ^1^ in 2022 [µSv/h] (Standard Error)	Mean Dose ^1^ in 2023 [µSv/h] (Standard Error)	Letters ^2^		Ecological Half-Life [y]
			2022	2023	
Bare land	0.04 (0.001)	0.03 (0.007)	C	A	2.41
Crop	0.06 (0.001)	0.03 (0.02)	A	A	1.00
Forest	0.05(0.007)	0.03 (0.02)	A B	A	1.36
Grassland	0.05 (0.003)	0.03 (0.02)	B	A	1.36
Rice paddy	0.05 (0.012)	0.03 (0.02)	ABC	A	1.36
Urban	0.04 (0.0007)	0.02 (0.02)	C	B	1.00
Solar Panel		0.03 (0.02)	A	B	-

^1^ Subtracted from the background level of 0.04 µSv/h. ^2^ Levels not connected by the same letter are significantly different (*p* < 0.05).

## Data Availability

Data are contained within the article.

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
