# Peer review of "Low-Cost Sensor Deployment on a Public Minibus in Fukushima Prefecture"

_sensors, 2024, doi:10.3390/s24051375_

Round 1
Reviewer 1 Report
Comments and Suggestions for Authors
The reduction of the ambient radiation dose in Fukushima is demonstrated over a period of one year using a dosemeter deployed on a public bus.
It difficult to fully understand the manuscript due to the missing pieces of information regarding the methods implemented in the present work. The following issues must be all addressed to arrive at a proper recommendation for publishing this work.
1. The manuscript did not provide the coordinates where the measurements took place nor the distance of the bus route. This may lead to misinterpret the results. At least the start and stop points of the bus should be known. Moreover, maps in Figs. 2 and 3 should include the GPS coordinates.
2. Fig. 1 should show clearly where the dosemeter is located in the bus. Layout of the bus and dosemeter location is more informative than a photo of the bus.
3. The statistical analysis in the manuscript is not sufficient. For example, the uncertainties of the measured quantities in all tables are not given.
4. What do the squares in Figs.2 and 3 mean? Measuring point?
5. “Ubrain” in the legend of Fig. 2 is incorrect.
6. The style of writing the quantities is inconsistent. Sometimes there is no space between the number and the unit (for instance, line 162).
7. Notation of writing the radioisotopes is incorrect.
8. References style is completely unprofessional. Too many references to broken links. Style of the references does not match the standards. Almost, every reference has some flaws. Majority of references are public information rather then academic.
9. Some words are not appropriate (for example “lengthiest” in line 336) -> longest.
10. An elaboration of the reasons behind the variation of ecological half-life among different classifications of land is needed.
Author Response
"Please see the attachment."

Reviewer 2 Report
Comments and Suggestions for Authors
It has been a long time since I have read such well written, well organized, and technically well composed paper. It was a pleasure reading it, and I shall gladly recommend it as a literature reference to my colleagues and students after it is published.
I do not find any issues, neither concerning the composition of paper, nor technical problems. I do have one question, thou, was the sensor cleaned periodically or just kept as is? There is a problem of dust accumulation and therefore radionuclides accumulation on the surface of sensors.
In all, the idea and the organization of the measurements, using QGIS i.e., and other technical details are well designed.
Author Response
"Please see the attachment."

Reviewer 3 Report
Comments and Suggestions for Authors
This paper analyzes data to observe the year-long decline in ambient radiation doses in reconstructed areas of Fukushima prefecture after the nuclear accident of 2011, using a novel mobile monitoring system installed on a public bus.
The paper is clear. Results are well presented and discussed.
I have only few minor comments:
- line 67: "select" --> "selected"
- line 154-155: please check the sentence
- line 165: travelling at 40km/h in 30 seconds the covered distance is 333 m; why authors say 670 m? Please clarify
Author Response
"Please see the attachment."

Reviewer 4 Report
Comments and Suggestions for Authors
My major concern with this paper refers to the lack of discussion/consideration on the measurements performed on "road" and not on land. The dose rate on road is significantly lower than the average in the areas and lower than in field or forests. This has to be discussed, including the evolution of the areas from 2022 to 2023 with regard to the occupancy due to the lifting of order of evacuation.
In this context, the conclusion on ecological half life needs to be considered cautiously, and better discussed. You can't generalise the results observed on the road to all the areas.
In addition, references to other citizen monitoring should be welcome (SafeCast and D Shuttle notably).
The section from line 286 to line 291 is not correct. Please refer to ICRP Publication 146 from 2020 and avoid to mention dose limit which is not applicable in such a situation.
The use of the background is not appropriated as it is. The value of 0.04 µSv per h was an average value for all Japan. Please be careful and probably this is not directly useful to substract for your analysis.
The meaning of ecological half life is really sensitive, and I would suggest to avoid conclusion on this issue: we may have the feeling that in 2024 the dose rate will still decrease with the same ratio. This is not relevant. You need to better integrate the decontamination performed on the road in the last few years and the evolution of the municipalities of concern.
Author Response
"Please see the attachment."

Round 2
Reviewer 1 Report
Comments and Suggestions for Authors
The authors have addressed most of my comments. However, the GPS coordinates in Fig. 2 and 3 of the revised version are not correct. The authors are advised to follow the high standard of producing these figures. An example to do so can be found in Fig. 1 and 2 of Ref. 23.
Reviewer 4 Report
Comments and Suggestions for Authors
The revised version is much more better integrating the various factors to explain the results.
